# Synthesis of Three-Dimensional Carbon Nanosheets and Its Flux Pinning Mechanisms in C-Doped MgB_2_ Superconductors

**DOI:** 10.3390/ma15217530

**Published:** 2022-10-27

**Authors:** Qian Zhao, Yun Chen, Baojun Qin, Chunhao Hu, Guoqing Xia, Liang Hao, Xuecheng Ping

**Affiliations:** Tianjin Key Laboratory of Integrated Design and On-Line Monitoring for Light Industry & Food Machinery and Equipment, School of Mechanical Engineering, Tianjin University of Science & Technology, Tianjin 300222, China

**Keywords:** superconductor, pinning center, carbon nanosheets, flux pining, connectivity

## Abstract

Three-dimensional carbon nanosheets (3D-CNS) were synthesized by salt template spray-drying method in order to solve the agglomeration of 2D nanocarbon by a traditional mixing method. MgB_2_ bulks doped with 3D-CNS with molar ratio composition of MgB_2−x_(3D-CNS)_x_ (x = 0, 0.1 and 0.2) have been prepared by in situ sintering process. The microstructure, critical current density and flux pinning of the sintered samples have been investigated. Differing from the structure in previous studies, the 3D-CNS doping is more efficient for the refinement of the MgB_2_ grains due to the 3D network structures. The results clearly show that more active C releasing from 3D-CNS at high temperature can provide effective flux pinning centers by the substitution of C for B in MgB_2_ lattice. Furthermore, the lattice distortion and increased grain boundaries should be responsible for the enhancement of critical current density (*J_c_*) at high magnetic fields as well as the increased irreversible magnetic field (*H_irr_*). However, the positive action in *J_c_* at low field has been extremely offset by the concentration of impurities at MgB_2_ grain boundaries such as released extra C without substitution and MgO, which is considered to further deteriorate the grain connectivity.

## 1. Introduction

High-temperature superconductivity (HTS), such as BKBO superconductor, can greatly save application costs due to their specific high critical transition temperature (*T_c_*) [1]. The intermetallic compound MgB_2_ has also caught worldwide attention due to its unusually high *T_c_* at 39 K [2]. The intrinsic properties such as simple crystal structure, low material cost and the operating temperature between 10 and 35 K offer the possibility of wide engineering applications [3]. Compared with copper oxide superconductors, although MgB_2_ does not have “weak link” grain boundaries, Mg powder is prone to volatilization and oxidation in the preparation process of traditional solid-state sintering, which inevitably affects the grain connectivity of MgB_2_, thus reducing the current carrying performance at low field. Moreover, the limited grain boundaries, as the main source of flux pinning in pure MgB_2_, make the critical current density (*J_c_*) decrease rapidly with the increase of magnetic field due to the lack of effective flux pinning center at medium-high field [4]. In the past years, enormous efforts have been devoted to obtaining a high-performance MgB_2_ superconductor by improving sintering technology or chemical doping. The high-density MgB_2_ bulk synthesized by Dadiel et al. using the spark plasma sintering (SPS) method greatly improved the *J_c_* of the sample under low magnetic field [5]. Carbon doping as a mainstream additive has been widely used in various superconductors. For instance, Multi-Wall Carbon Nano-Tubes (MWCNTs) adding to YBCO-123, MgB_2_ and Gd-123 have been tried by many researchers previously [6,7,8]. As well silicon carbide (SiC) [9], nanocarbon [10], carbon nanotube (CNT) [11] and carbon-containing compounds has been identified to be the most effective method for enhancing *J_c_* originating from the substitution of C for B in the MgB_2_ lattice [12,13,14,15,16]. Among these, graphene is selected as a new dopant for MgB_2_ because of the single-layered sheet structures. The more flux pinning centers in doped MgB_2_ samples may attribute to (1) C substitutions for B with an increase in intraband scattering, and (2) more lattice defects by the mismatch in thermal expansion between graphene and the MgB_2_ matrix [17,18]. However, it is not feasible to enhance the flux pinning only by increasing the content of graphene. The previous study indicated that it was a great challenge for the achievement of homogeneous doping due to the entanglement between graphene sheets, which performed current obstruction and thus to the reduction of *J_c_* in high fields [19]. In order to deal with these problems, some research groups attempted to use carbohydrate as carbon source to achieve the homogenous carbon substitution by the decomposition of the compounds during the heating process [20,21]. However, there was no definitive control during the sintering process and oxygen from hydrocarbon could produce large-scale MgO impurities. Recently, a more efficient method called carbon-coatings for boron has successfully achieved the uniform dispersion [22,23]. The carbon with the thickness of 1-2 nm has uniformly encapsulated the boron powder and achieved the high field property enhancement. The similar in situ coating techniques such as CNT-and amorphous carbon-coated Mg [24], graphene-coated boron [25], carbon capsulated Ni nanoparticles [26], and carbon-encapsulated Fe nanospheres [27] have been applied in doping of MgB_2_ samples to guarantee the uniform distribution of C and a positive effect in enhancing the *J_c_* properties. Nevertheless, the characterization of the formed shell-core encapsulated structure has not been well illustrated so far due to the complicated synthesis methods. In order to obtain uniform dispersion and stable structure of carbon materials in doped MgB_2_, an in situ synthesis of 3D carbon nanosheets via a spray drying method is investigated in the present work and further doped into MgB_2_ bulk to overcome the irreversible aggregation or re-stacking of 2D graphene in matrix. The spray drying method has the characteristics of simple operation, high efficiency and mass production of materials. Until now, this technique has been successfully applied in various fields. Zhang et al. have fabricated metal matrix composites reinforced with 3D graphene-like network [28]. Qin et al. have developed SnSb-in-plane nanoconfined 3D N-doped porous graphene network for sodium ion battery anodes [29]. Liu et al. have reported 3D carbon nanosheets anchored with Fe_3_O_4_ nanoparticles for electromagnetic wave absorption [30]. To our best knowledge, this fabrication method has not been applied in doping of MgB_2_ superconductor, and the first attempt is expected to reach two goals: (1) The agglomeration of the impurities is solved by an effective solution in the composite. (2) The connectivity among MgB_2_ grains is improved by 3D carbon network structure and highly efficient substitution of C.

## 2. Experimental Procedure

This process mainly comprises two steps, including the preparation of the precursor 3D carbon nanosheets and fabrication of doped MgB_2_ bulks by mixing the composites powder with Mg and B. The detailed process is as follows:

(i). 52.0 g of NaCl and 2.666 g of C_6_H_12_O_6_ were dissolved in 400 mL deionized water stirring for 4 h to obtain a homogeneous mixed solution. The solution was spray-dried to obtain the precursor powder under the inlet temperature of 150 °C. The powder was then kept in a quartz boat and calcined in flowing Ar at 600 °C for 2 h for carbonization. Subsequently, the as-prepared composite powders were washed with deionized water several times to remove NaCl, and finally precursor powder was obtained after drying at 80 °C. 

(ii). The 3D carbon nanosheets (3D-CNS) composites powder was thoroughly mixed with amorphous boron powder (99% purity, ~25 mm) and magnesium powder (99.5% purity, ~100 mm), and then mechanically pressed into cylindrical pellets using a pressure of 3 MPa. In order to obtain thermo kinetic information during the sintering, the selected samples were heated in the TGA/DSC1 thermal analyzer under the protection of argon atmosphere with a heating rate of 5 °C/min. The stoichiometric ratio and detailed experimental parameters had been shown in Table 1. To facilitate the discussion below, the samples were denoted as 3D-CNS0, 3D-CNS0.1 and 3D-CNS0.2, respectively, according to the doping level of 3D-CNS in MgB_2_. The phase identification was carried out using X-ray diffraction (XRD). Raman spectroscopy (Renishaw in Via Raman Microscope, Woodchester, UK) at 532 nm characterize the graphene-like powder. Scanning electron microscopy (SEM) and transmission electron microscopy (TEM) were utilized to the microstructural observation of the precursor powder and doped MgB_2_ samples. Superconducting properties were measured by a physical property measurement system (Quantum Design MPMS-9, San Diego, CA, USA). Before the magnetic measurement, the cylindrical pellets were polished into a cuboid sample with the size of 1 × 2 × 4 mm^3^. The magnetic *J_c_* values were calculated from the width of magnetization hysteresis loops based on Bean model [31]. The expression of the Bean model is *J_c_* = 20 Δ*M*/[*abh*^2^(1 − *h*/3*b*)], where Δ*M* is the magnetic hysteresis loop width in external magnetic field *H*, and the unit is emu. *A* represents the long side of the cuboid sample, and *h* and *b* represent the short side with *h* < *b* (unit: mm).

## 3. Results and Discussion

The synthesis process of the 3D-CNS precursor powders via a spray drying method is illustrated in Figure 1. In this procedure, the solution (NaCl + C_6_H_12_O_6_) was sprayed to small droplets by a constant output atomizer, and subsequently the water was removed rapidly with hot air. The NaCl crystallized into small cubes coated with an ultrathin C_6_H_12_O_6_ film, thus yielding the micrometer-sized spheres constructed NaCl + carbon composites after calcinating at high temperatures. Finally, the 3D carbon nanosheets architecture was obtained after removing the NaCl template by distilled water. 

Figure 2a shows the XRD pattern of the 3D-CNS, the diffraction peaks at 26.228°and 44.363°assigned to the (002) and (110) planes of carbon reflect the completely glucose carbonization. In addition, the peaks of carbon in the pattern are broader and weaker than the standard JCPDS card, indicating the imperfect crystallinity of three-dimensional carbon with structural defects. The Raman spectrum of 3D-CNS is displayed in Figure 2b. It is clearly that there are two characteristic peaks at about 1350 and 1596 cm^−1^, corresponding to the *D* band and *G* band, respectively. The *G* band originates from the stretching motion of *sp*_2_ carbon pairs in both rings and chains, while the *D* band arises from defects in the hexagonal *sp*_2_ carbon network or the finite particle-size effect [32]. It is obvious that the *D* band associated with disordered or defective carbon is broader in the pattern, and the calculated intensity ratio of *D* band to *G* band (I*_D_*/I*_G_* = 1.01) further demonstrating the existence of defects and disorders in 3D-CNS precursor. The Raman analysis is in accordance with the XRD results, and the characterization of the microstructure will be further identified by SEM and TEM in below. 

Figure 3 shows the representative SEM and TEM images of 3D-CNS. From Figure 3a and the elemental mapping images, it can be clearly seen that the precursor powder is composed of interconnected carbon nanosheets with apparent spherical 3D structure, which is derive from the shape of the NaCl cubes. Combined with the element mappings and EDS patterns in Figure 3d, the tiny oxygen existed except for C as main element demonstrates that the CO_2_ molecules from the oxidization of the 3D-CNS during the preparing process have been partly adsorbed on the surface of the powders. In Figure 3b, some multiple rings in the three-dimensional porous structure have been observed except for the hexatomic ring structure. In addition to the six-membered ring structure with *sp*_2_ hybrid carbon atom rearrangement, other multiple rings appear, which introduce intrinsic defects. On the basis of the TEM observations in Figure 3c, the multiple carbon nanosheets presented in the powder is well consistent with the SEM observations.

To well understand the transformation process by the thermal analysis apparatus, the recorded DTA thermal signals as a function of temperatures are illustrated in Figure 4. Briefly, the DTA curve of pure MgB_2_ bulk contains three thermal peaks, which is corresponding to solid–solid reaction (Peak 1), solid–liquid reaction (Peak 3) and the melting of Mg at 650 °C (Peak 2). The similar procedure can also be found in the 3D-CNS doped samples. It is noted that the intensity of Peak 3 of the three samples is higher than that of Peak 1, indicating a rapid reaction between Mg and B under high temperature (700 °C). As for the heating temperature below 550 °C, the reaction is restricted by the slow diffusion rate of Mg atom at solid state. In other words, the 3D-CNS doping cannot essentially promote the reaction procedure in forming of MgB_2_ phase. However, it is noteworthy that the starting temperature of the solid-state reaction of the doped samples is about 25 °C forward than that of the pure sample. The results shows that the network structure of 3D-CNS benefits for providing more contact area for Mg and B atom and promotes the nucleation and crystallization of MgB_2_ during solid state diffusion. In contrast, the secondary sintering stage (Peak 3) after Mg melting is slightly delayed by 3D-CNS addition, attributing to the destroyed 3D network structure by releasing activated carbon into MgB_2_.

The XRD diffractograms of the bulks are shown in Figure 5. It can be clearly seen that MgB_2_ is the dominant phase with a small amount of MgO and carbon impurities in doped samples. The oxidation of Mg is inevitable during the sintering process, while the existence of carbon is expected to bring in effective carbon substitution for B. The volume fraction of MgO was evaluated from the peak intensity of the patterns by Rietveld method [33].

Accordingly, the volume fraction of the other phases (MgB_2_ and C) can also be obtained based on the above calculation and is shown in Figure 5b. It is obvious that the amount of MgB_2_ experience a visible decrease with the doping of 3D-CNS, which is in consistent with the DTA curve analysis. In order to evaluate the C substitution in MgB_2_, the local enlarged patterns of (002) and (110) peaks are displayed in Figure 5c. Compared to pure sample, a remarkable shift to high angle of (110) peak has been evidential to the substitution of C on B in 3D-CNS doped samples. The full width at half maximum (FWHM) values for (110) peak in Table 2 indicates that the MgB_2_ grain has small size and imperfect crystallinity by the concentration of the maintained carbon on grain boundaries. To confirm the level of C substitution, the values of y in the formula Mg(B_1−y_C_y_)_2_ could be calculated from the lattice parameters *c* and *a*, as y = 7.5 Δ(*c*/*a*) [34], where Δ(*c/a*) is the deviation of *c/a* from that of pure sample. The results could be found in Table 2. It suggested that the 3D-CNS has generated effective carbon substitution from 0.017 to 0.021 by the increase of doping level, which is comparable to the reported carbon dopants, for instance, y = 0.024 for carbon nanotubes and y = 0.025 for glycine [35,36]. Therefore, the lattice distortion derives from the carbon substitution should be advantageous for the flux pinning and the *J_c_* enhancement at high field. 

The SEM images of doped 3D-CNS samples are shown in Figure 6. The undoped sample reveals the typical hexagonal structures with sharp grain boundaries, as seen in Figure 6a,d. On the contrary, the doped samples exhibit visible microstructure that differs from the pure one. In Figure 6b,c, a large number of “boundaryless regions” without the formation of hexagonal regular grains can be observed, where the grains seem to be smaller and more compacted compared to pure sample. The grain refinement is further supported by the FWHM of the (110) peak in Table 2. Moreover, the existence of the holes in between MgB_2_ grains (red circle area) is distinguished in Figure 6e,f, which is possible originating from the partly evaporation of Mg due to the delay of the reaction between melt Mg and B by 3D-CNS doping. The detailed reaction process can be confirmed in DTA analysis. With respect to the visible MgB_2_ lamellar structure (red arrow), According to literature reports [37], MgB_2_ will preferentially grow in the (101) crystal planes, and the slowest growing planes is the (001). If the impurity phase plays the role of pinning, it will inhibit the growth in the (001) direction, indicating that the addition of 3D-CNS plays the role of flux pinning to a certain extent, and reduces the size of the grain in the C-axis direction. From the SEM observation, it is concluded that the 3D-CNS has introduced intra-grain pinning into MgB_2_ whereas producing weak links at the grain boundaries. A detailed discussion is necessary to clarify the effect of 3D-CNS on the electromagnetic properties of MgB_2_.

The temperature dependence of normalized ZFC magnetization for the prepared samples is illustrated in Figure 7. The sharp superconducting transition in pure sample indicates the high crystallinity and few defects in MgB_2_ grains, which is in accordance with the SEM observation. As for the doped samples, the critical transition temperatures (*T_c_*) value decreased to 36 K for 3D-CNS0.1, and subsequently to 33.76 K for 3D-CNS0.2. Both of them are definitely lower than that of the pure MgB_2_ sample (38.53 K). The result is confirmed by the previously research due to the extrinsic of carbon substitution, when carbon doping is introduced, the impurity scattering will be increased and the electron phonon coupling will be weakened, and the substitution of carbon for B-site will cause lattice distortion, which inducing the *σ* hole-band filling from the extra electron of carbon atom and lattice distortion in MgB_2_. Moreover, the large amount of small MgB_2_ grains with more defects and inferior crystallinity also contribute to the lower value of *T_c_* in doped sample. In consequence, the *T_c_* values exhibit a dramatical decline with the increase doping level of 3D-CNS. 

Figure 8b illustrates the measured *J_c_*-*H* characteristics of the undoped and 3D-CNS doped samples, in which the values of *J_c_* were calculated from the magnetic hysteresis loops shown in Figure 8a. It should be noticed that the *J_c_* curves for all doped samples experience an obvious decrease in low fields compared to the pure sample. It is known that the carbon from 3D-CNS could substitute the B site in MgB_2_ lattice structure. From the above analysis, it can be seen that the addition of 3D-CNS will make C atom enter the lattice of MgB_2_ to replace B atom, and the outermost electron of C atom is one more than that of B atom, which belongs to electron doping. Since the superconductivity of MgB_2_ relied on the holes inside the B planes, the introduced electrons by C atoms will decrease the concentration of the carriers, which will reduce the *J_c_* at low fields. In addition, the *J_c_* value in the low magnetic field is mainly related to grain connectivity. As for the released free carbon from 3D-CNS at high temperature, the unsubstituted carbon deposited at the grain boundary will weaken the grain connectivity of MgB_2_, thus resulting in the increase of weak links and the suppression of *J_c_* value in low field. It is worth noting that the doping of 2D graphene with the same content (10% mole percentage) will not reduce the *J_c_* under low field [38], which further indicates that the 3D-CNS structure defects synthesized in situ in this work make its thermal stability poor. It is precisely after the structure is destroyed at high temperature that more C is released to reduce the grain connectivity of the sample. On contrary, a significant improvement of *J_c_* at high field in doped sample reveals that the 3D-CNS can introduce more magnetic flux pinning centers and improve the critical current density value. Combined with the SEM image and *T_c_* analysis, it is concluded that the lattice distortion by C substitution of B and more grain boundaries all contribute to the *J_c_* enhancement. Accordingly, the irreversibility field *H_irr_* is raised from 4.5 T to 4.8 T as increasing the doping level, higher than that of undoped sample (*H_irr_* = 4.1 T). However, the advantage of the 3D-CNS doping is visibly weakened as the content of 3D-CNS increasing from 0.01 to 0.02. It can be seen that the *J_c_* value of 3D-CNS0.1 drops more dramatically at full fields than that of 3D-CNS0.2 sample. The larger concentration of impurity and greater flux penetration in the grain boundaries are thought to be the main reasons for the suppression of *J_c_* by the excessive doping level. 

In order to further interpret the pinning mechanism for the pure and doped samples, the field (*H*) dependence of the flux pinning force (*F_p_*) is illustrated in Figure 9. The flux pinning force is described by *F_p_* = *μ_*0*_J_c_*(*H*)*H*, where *μ*_0_ is the magnetic permeability in vacuum [39]. Compared to the doped sample, the peak area is found to be the largest one at 2 T for pure sample, which is strongly correlated with better intergranular connectivity without 3D-CNS doping in low field. On the contrary, the flux pinning force of 3D-CNS0.1 after 2 T and that of 3D-CNS0.2 after 3.2 T are both higher than that of pure MgB_2_, corresponding to the *J**_c_* results in Figure 8. The specific magnetic flux pinning mechanism is further analyzed as follows:

To compare the behavior of the samples, the normalized flux pinning force (*f_p_* = *F_p_/F_pmax_*) as a function of the normalized field (*h = H/H_irr_*) has been plotted in Figure 10a based on Hughes model (*f_p_* = *Ah^m^*(1 − *h*)*^n^*) [40]. As for MgB_2_ superconductor, the pinning types are mostly grain boundary pinning and point pinning, so their characteristic values are listed in Table 3 for reference. It can be seen from Table 3 that the fitting result of the samples by Hughes model is remarkably consistent and with the correlation coefficient reaching above 0.99. It is noticed that the *m*, *n* and *h_peak_* values of the pure and doped samples have obvious deviation from the theoretical values with the respect of point pinning and surface pinning. Some research groups have also observed the similar phenomena but without given a specific explanation. In this study, another Higuchi model [41] is employed to further determine the flux pinning mechanism.

The normalized field *h* dependence of the normalized flux pinning force *f* was illustrated in Figure 10b with magnetic field for each sample at 20 K. According to Higuchi pinning model, normalized pinning force density *f* = *F_p_/F_pmax_* is often scaled with *h* = *H/H_max_*, instead of *h = H/H_irr_*. The formula curves of grain boundary pinning and point pinning are also shown in the figure, and the expressions are, respectively:$$\left\{ \begin{array}{ll} f(h)=2.25h{\left(1-h/3\right)}^{2} & \left(1\right)\\ f(h)=1.5625h^{1/2}{\left(1-h/5\right)}^{2} & \left(2\right) \end{array}\right.$$

In Figure 10b, it is shown that the curves of 3D-CNS doped samples in the low field almost coincide with the point pinning curve, indicating that the point pinning mechanism plays a major role in the low field. As for the high magnetic field, the curve gradually approaches the grain boundary pinning curve and even exceeds the curve for 3D-CNS0.2 sample, which indicates the pinning type changing from point pinning to surface pinning. According to the previous analysis, it is deduced that the more C atoms enter into the MgB_2_ lattice the less anisotropy in grains, and thus making the magnetic flux pinning density curve move toward the surface pinning curve.

## 4. Conclusions

We have analyzed the effects of grain connectivity, microstructure and flux pinning on critical current property of bulk MgB_2_ materials by 3D-CNS doping. The present study shows that the doping of three-dimensional carbon nanosheets is a promising way to effectively improve the current carrying performance of MgB_2_ in high field. The contribution of the 3D-CNS doping is the high efficiency substitution of C for B and the refinement of MgB_2_ grains by the three-dimensional carbon network. The lattice distortion, nano C and the increase of grain boundaries play a positive role in the enhancement of flux pinning. Nevertheless, there is no evidence of significantly improved grain connectivity with the addition of 3D-CNS. The large aggregation of free carbon from the damage of 3D-CNS structure at high temperature has created harmful weak links among MgB_2_ grains. In summary, the doping of 3D-CNS is proven to perform more efficiently pinning centers than conventional carbohydrate and graphene doping with relatively stable structure. Further research should focus on the collaboration of metal and 3D-CNS to improve the grain connectivity by liquid-assisted sintering mechanism.

## Figures and Tables

**Figure 1 materials-15-07530-f001:**
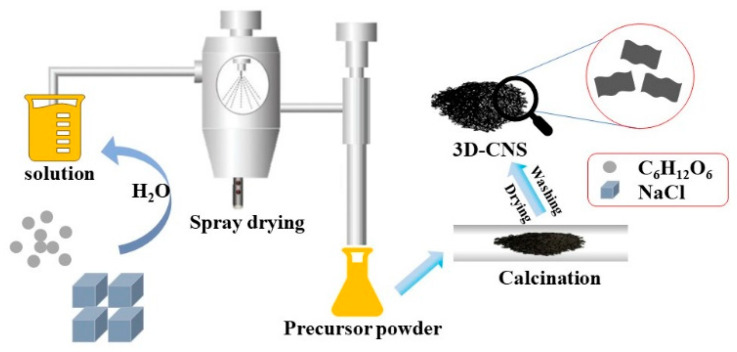
Schematic illustration of the synthesis process of the 3D-CNS precursor powder by spray drying.

**Figure 2 materials-15-07530-f002:**
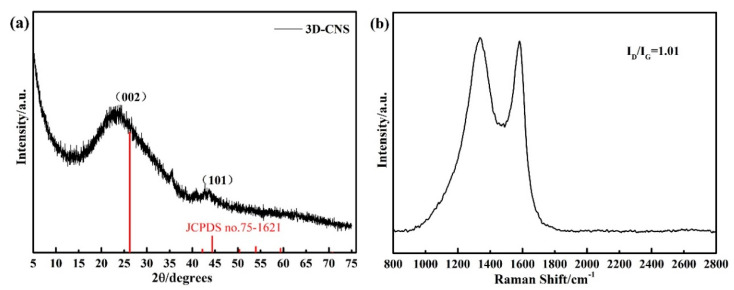
(**a**) XRD pattern and (**b**) Raman spectrum of 3D-CNS.

**Figure 3 materials-15-07530-f003:**
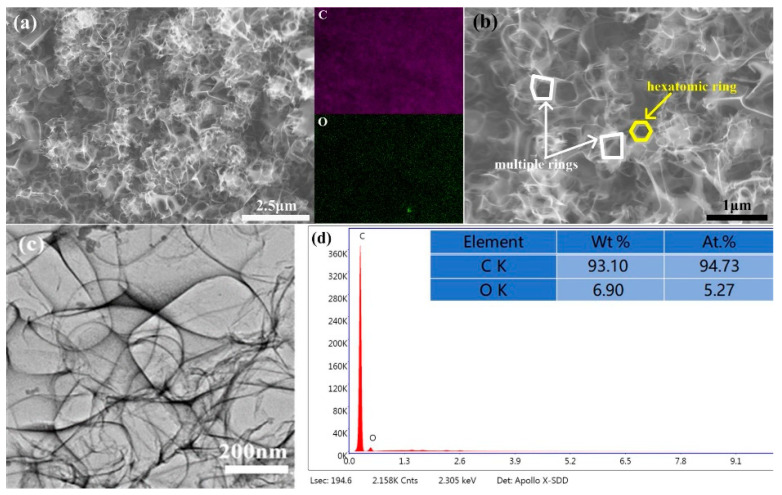
(**a**,**b**) SEM, (**c**) TEM and (**d**) Elemental surface scan of 3D-CNS.

**Figure 4 materials-15-07530-f004:**
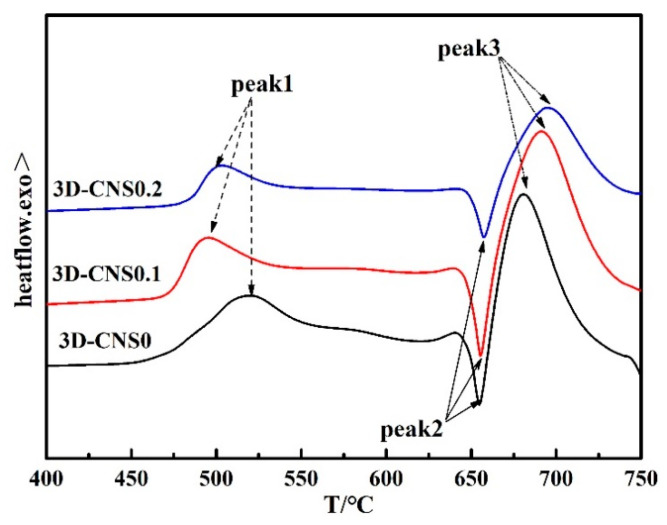
The measured DTA curves for pure and 3D-CNS doped MgB_2_ samples.

**Figure 5 materials-15-07530-f005:**
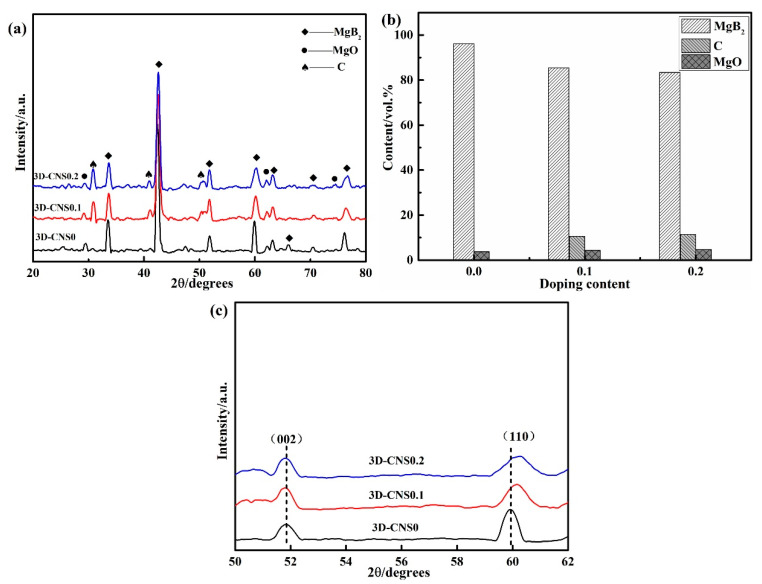
(**a**) XRD patterns (**b**) volume fraction of each phase and (**c**) local magnification of 3D-CNS doped samples.

**Figure 6 materials-15-07530-f006:**
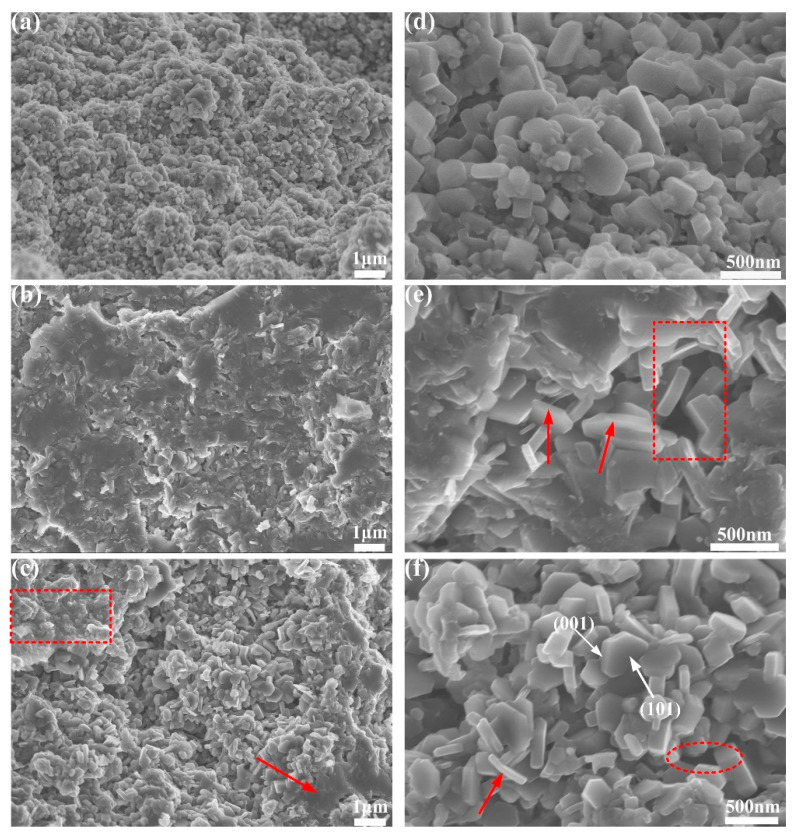
SEM images of doped 3D-CNS samples at low (**a**–**c**) and high magnification (**d**–**f**): (**a**,**d**) 3D-CNS0, (**b**,**e**) 3D-CNS0.1, (**c**,**f**) 3D-CNS0.2.

**Figure 7 materials-15-07530-f007:**
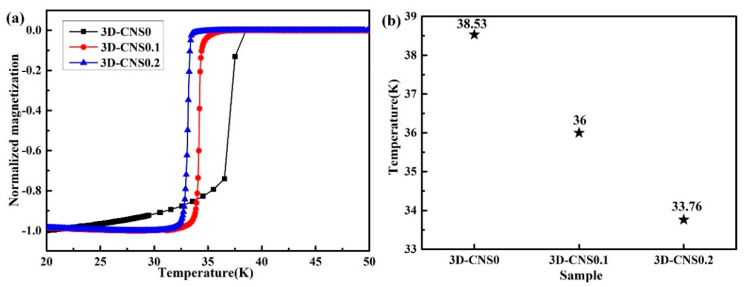
(**a**) Temperature dependences of normalized zero field cooled (ZFC) magnetization (**b**) *T_c_* comparison of doped 3D-CNS samples.

**Figure 8 materials-15-07530-f008:**
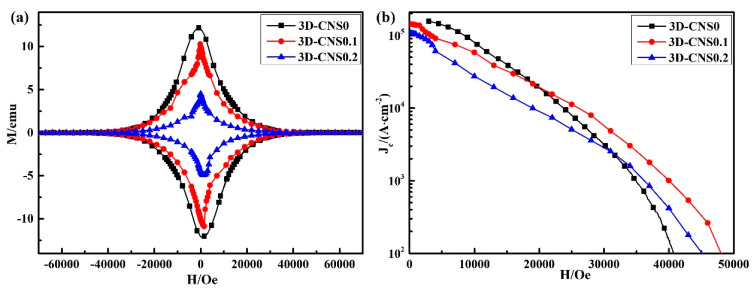
(**a**) Magnetic hysteresis loops (**b**) critical current density (*J_c_*) as a function of magnetic field at 20 K for undoped and 3D-CNS doped samples.

**Figure 9 materials-15-07530-f009:**
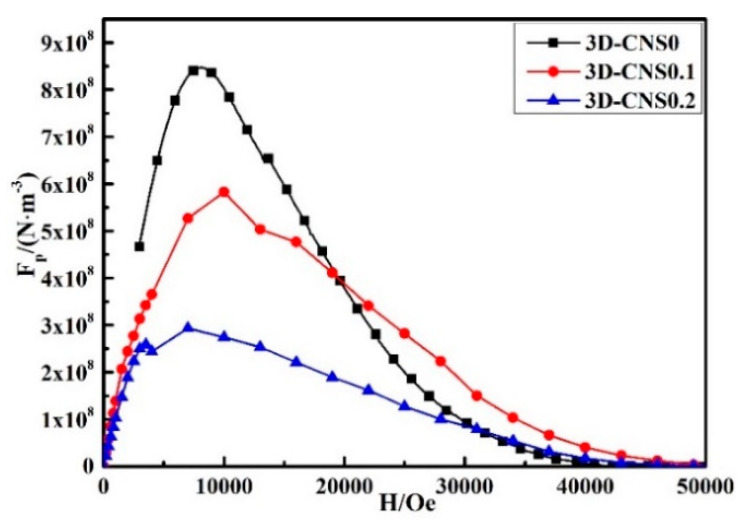
Flux pinning force curves of undoped and 3D-CNS doped samples.

**Figure 10 materials-15-07530-f010:**
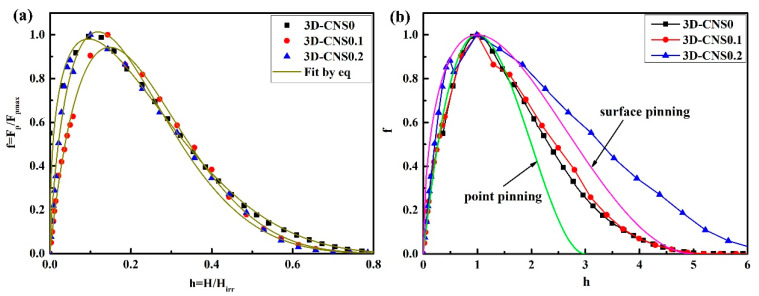
(**a**) Fitting of Hughes pinning model and (**b**) variation curve of normalized pinning force density with magnetic field for composite samples doped with different 3D-CNS contents.

**Table 1 materials-15-07530-t001:** Overview of parameters for preparing composite samples.

Sample	*n*(Mg): *n*(B): *n*(3D-CNS)	Temperature/°C	Duration/h	Size/mm
3D-CNS0	1:2:0	750	0.5	Φ5 × 1.5
3D-CNS0.1	1:1.9:0.1			
3D-CNS0.2	1:1.8:0.2			

**Table 2 materials-15-07530-t002:** Comparison of lattice parameters, carbon doping level, characteristic peak position and FWHM of doped 3D-CNS composite samples.

Sample	2*θ* of (002) Peak	2*θ* of (110) Peak	FWHM of (002) Peak	FWHM of (110) Peak	*a*/Å	*c*/Å	y in Mg(B_1−y_C_y_)_2_
3D-CNS0	51.879	59.938	0.324	0.266	3.0859	3.5230	/
3D-CNS0.1	51.859	60.143	0.318	0.698	3.0777	3.5192	0.017
3D-CNS0.2	51.852	60.289	0.341	0.854	3.0581	3.5255	0.021

**Table 3 materials-15-07530-t003:** Comparison of fitting values of Hughes model.

Sample	*m*	*n*	*h* _peak_	R^2^
Point pinning	1	2	0.33	/
Surface pinning	0.5	2	0.2	/
3D-CNS0	0.40	3.78	0.11	0.9913
3D-CNS0.1	0.95	5.45	0.15	0.9961
3D-CNS0.2	0.73	5.40	0.12	0.9856

## Data Availability

Data sharing not applicable.

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
