# Peer review of "Synthesis of Three-Dimensional Carbon Nanosheets and Its Flux Pinning Mechanisms in C-Doped MgB2 Superconductors"

_materials, 2022, doi:10.3390/ma15217530_

Round 1
Reviewer 1 Report
The work is interesting, but needs minor corrections and minor bugs to be corrected nevertheless:
- MgB2 has a critical temperature of 39 K, but there are also other superconducting perovskites with a similar critical temperature, which are also widely used, including BKBO. It is worth the authors mention it: https://doi.org/10.1016/j.actamat.2021.117437
- I also miss information in the introduction about the Spark Plasma Sintering Method for obtaining MgB2: 10.3390/ma14237395
- Maybe it is worth adding information that CNTs were also added to other superconductors, e.g. YBCO: 10.1016/j.molstruc.2020.128089
- Tables and Figures should match the Materials MDPI template, eg Fig. 2 and Tab. 1 and others. It should be like Fig. 1.
- In addition, the year in the upper footer is 2021, and now we have the year 2022
- sp2 orbitals - it should be "sp" in italics and "2" in subscript
- "D" and "G" should be in italics, they are variables.
- Please explain the vertical axis in Fig. 4.
- Fig. 8. Jc/A*cm-2 - this is wrong, it should be Jc[A*cm-2]. This suggests dividing by units. Please put units in square brackets.
- Fig. 9 - please explain the markings in the figure caption
Reviewer 2 Report
The authors synthesize MgB2-C composite by mixing powders of Mg, B and carbon nanosheets and subsequent solid-phase and liquid-phase synthesis. As a result, the properties of 3 samples with a fraction of carbon nanosheets of 0, 0.1, 0.2 were obtained and studied. It is shown that some carbon atoms replace boron atoms in the MgB2 structure, which apparently leads to a decrease in Tc. This, as well as the remaining carbon, has an effect on the capture of magnetic flux in the resulting superconductor, while the effect of the addition of carbon on the critical current is positive in some fields. The topic of the work is relevant, the material as a whole is conveyed clearly. The article can be accepted subject to minor comments.
1. The abstract mentions 3D-CNS doping with a composition of MgB 2-x C x (x=0, 0.1 and 0.2). This is confusing because the notation MgB2-xC x implies doping at the atomic level. This place needs to be changed, refusing to write MgB2-xC x.
2. The calculation of Jc using the Bean model implies the use of a hysteresis loop. It would be helpful to demonstrate this loop. For example, because the magnetization hysteresis can demonstrate impacts of second phase.
3. Quantitative data on Jc calculated using Bean's model require the choice of the critical current radius parameter. This choice can be related to both sample size and microstructure parameters (see, for example, [Gokhfeld, D.M. The Circulation Radius and Critical Current Density in Type II Superconductors. Tech. Phys. Lett. 2019, 45, 1–3, doi :10.1134/S1063785019010243.], [Altin, E.; Gokhfeld, D. M.; Komogortsev, S. V.; Altin, S.; Yakinci, M. E. Hysteresis Loops of MgB2 + Co Composite Tapes. J. Mater. Sci. Mater. Electron. 2013, 24, 1341–1347, doi:10.1007/s10854-012-0931-2.]). Information on the choice of circulation radius must be provided.
Reviewer 3 Report
The authors first prepared 3D carbon nanosheets then studied their effect on flux pinning of MgB2. Overall, the text is well structured. The results appeared correct and interesting. The work can be accepted after minor revisions:
1) There are some minor grammatical and typo mistakes that should be corrected.
2) The novelty of the work is not well highlighted.
3) Why do the authors used argon atmosphere during preparation of the C- doped MgB2.
4) More details in the experimental section should be provided such as the sintering temperature, rate °/C, the used furnace, the pelletization etc.
4) The discussions should be improved. More comparison between the results of this work and those previsiouly reported for the same composition should be provided.
